# Human Pluripotent Stem Cell-Derived Neural Progenitor Cells Promote Retinal Ganglion Cell Survival and Axon Recovery in an Optic Nerve Compression Animal Model

**DOI:** 10.3390/ijms222212529

**Published:** 2021-11-20

**Authors:** Mira Park, Hyun-Mun Kim, Hyun-Ah Shin, Seung-Hyun Lee, Dong-Youn Hwang, Helen Lew

**Affiliations:** 1Department of Ophthalmology, CHA Bundang Medical Center, CHA University, Seongnam 13496, Gyeonggi-do, Korea; hoohoo9979@gmail.com (M.P.); sha9547@naver.com (H.-A.S.); lsh1013@chamc.co.kr (S.-H.L.); 2Department of Biomedical Science, CHA University, Seongnam 13488, Gyeonggi-do, Korea; kyn0708@naver.com; 3Department of Microbiology, School of Medicine, CHA University, Seongnam 13488, Gyeonggi-do, Korea

**Keywords:** human pluripotent stem cell-derived neuronal progenital cells (NPCs), human placenta-derived mesenchymal stem cells (PSCs), optic nerve compression model, neuroprotection, ganglion cell survival, axon recovery, microglial engagement, anti-inflammation

## Abstract

Human pluripotent stem cell-derived neural progenitor cells (NPCs) have the potential to recover from nerve injury. We previously reported that human placenta-derived mesenchymal stem cells (PSCs) have neuroprotective effects. To evaluate the potential benefit of NPCs, we compared them to PSCs using R28 cells under hypoxic conditions and a rat model of optic nerve injury. NPCs and PSCs (2 × 106 cells) were injected into the subtenon space. After 1, 2, and 4 weeks, we examined changes in target proteins in the retina and optic nerve. NPCs significantly induced vascular endothelial growth factor (Vegf) compared to age-matched shams and PSC groups at 2 weeks; they also induced neurofilaments in the retina compared to the sham group at 4 weeks. In addition, the expression of brain-derived neurotrophic factor (Bdnf) was high in the retina in the NPC group at 2 weeks, while expression in the optic nerve was high in both the NPC and PSC groups. The low expression of ionized calcium-binding adapter molecule 1 (Iba1) in the retina had recovered at 2 weeks after NPC injection and at 4 weeks after PSC injection. The expression of the inflammatory protein NLR family, pyrin domain containing 3 (Nlrp3) was significantly reduced at 1 week, and that of tumor necrosis factor-α (Tnf-α) in the optic nerves of the NPC group was lower at 2 weeks. Regarding retinal ganglion cells, the expressions of Brn3a and Tuj1 in the retina were enhanced in the NPC group compared to sham controls at 4 weeks. NPC injections increased Gap43 expression from 2 weeks and reduced Iba1 expression in the optic nerves during the recovery period. In addition, R28 cells exposed to hypoxic conditions showed increased cell survival when cocultured with NPCs compared to PSCs. Both Wnt/β-catenin signaling and increased Nf-ĸb could contribute to the rescue of damaged retinal ganglion cells via upregulation of neuroprotective factors, microglial engagement, and anti-inflammatory regulation by NPCs. This study suggests that NPCs could be useful for the cellular treatment of various optic neuropathies, together with cell therapy using mesenchymal stem cells.

## 1. Introduction

Because there is no effective therapy for irreversible damage to the optic nerve, many studies have attempted to improve the essential regenerative capacity of retinal ganglion cells (RGCs) [1]. Macrophage-activating factors and zymosan promote axon regrowth after optic nerve damage [2,3,4,5]. Changes in the inherent renewal capacity of RGCs may be caused by deficiencies of phosphatase and tensin homolog (PTEN) [6,7]. The combination of inflammatory induction through injection of zymosan, PTEN deficiency, and promotion of intracellular cyclic adenosine monophosphate (cAMP) can help restore the optic nerve [8,9]. However, approved treatments are difficult to use in clinical trials.

For incurable eye diseases, embryonic stem cells (ESCs), limbal stem cells, retinal pigment epithelial cells, and mesenchymal stem cells (MSCs) are used for cell therapy as part of regenerative medicine [10]. MSCs attract immune cells by releasing immune modulators, mediators, and chemokines [11]. In addition, they have neuroprotective effects by secreting elements via paracrine action [11,12]. In numerous studies that used animal models of glaucoma, the survival rate of RGCs increased when MSCs were injected intravitreally [13,14,15,16]. In one study of an ischemic model, the number of RGCs and expressions of BDNF, ciliary neurotrophic factor (CNTF), and fibroblast growth factor (bFGF) after MSC injection were increased [17]. Several studies have also identified the therapeutic effects of MSCs in ischemic models [18,19,20]. In addition to MSCs derived from bone marrow, those derived from human umbilical cord blood, dental pulp, and placenta show therapeutic effects in terms of generative induction and axonal growth of damaged optic nerves [1,21]. 

We previously reported that the regulation of hypoxia-inducible factor 1-alpha (Hif-1α) and growth-associated protein 43 (GAP43) of MSCs derived from human placenta (PSCs) promote axon survival in an optic nerve compression (ONC) model [22]. In addition, regulation of the NF-κb pathway plays an important role in the regulation of target proteins in PSCs [23]. Another study demonstrated the possibility of using hypoxia-preconditioned PSCs (HPPCs) in cell therapy for optic nerve damage by examining the effects of regenerated nerves from HPPCs on R28 cells and in an animal model of optic nerve damage [1]. 

Many different molecules help regulate axon regeneration after optic nerve damage including growth factors such as Wnt, CNTF, BDNF, and semaphorins; growth-inhibitory transcription factors such as Kruppel-like factor 4 (KLF4); and essential signaling mediators such as Signal transducer and activator of transcription 3 (STAT3) [24]. The mechanisms by which Wnt signaling promotes axon regeneration may include induction of these axon growth-promoting genes; Wnt signaling may also directly control growth cone remodeling by changing microtubule stability during axonal growth [25].

Recently, neural progenitor cells (NPCs) have been found to be useful as a new cell therapy platform for various diseases [26]. In this context, we investigated the safety and clinical efficacy of human pluripotent stem cell-derived NPCs in an ONC model. 

## 2. Results

### 2.1. Characterization of Human Pluripotent Stem Cell-Derived Neuronal Progenitor Cells (NPCs)

CHA15 human ESCs were differentiated into NPCs by treatments with 5 μM PKCβ inhibitor, and 1 μM DMH1 in the medium consisting of DMEM/F12, 10 μg/mL human insulin, 9 μg/mL transferrin, and 14 ng/mL selenite (Figure 1A). Expanded NPCs at passage 1 were positive for two representative NPC markers, SOX1 (~90%) and PAX6 (~75.6%), but negative for a typical neural crest stem cell marker, P75 (~0%) (Figure 1B). When further differentiated into mature neurons, the NPCs generated both early and late neuronal markers, TUJ1 and MAP2, respectively (Figure 1C).

### 2.2. NPCs Reduce Cell Apoptosis and Regulate Target Proteins

To assess the recovery function of NPCs, we performed cell viability tests. When cocultured with NPCs or PSCs, the viability of damaged R28 cells recovered by 48% and 7% more, respectively, than under hypoxic conditions (Figure 2A). In addition, NPCs regulated cleaved caspase-3 activity and Bcl-2 protein expression during apoptosis (Figure 2B). We also found that Hif-1α cocultured with PSCs showed less hypoxic damage. In contrast to Hif-1α, decreased expressions of neurofilaments(Nf), Gap43, NeuN, and Gfap under hypoxic conditions were significantly recovered by coculturing them with NPCs (Figure 2C).

### 2.3. Changes in Neurogenic Marker Expression in the Retina after Injection of NPCs or PSCs in the Optic Nerve Compression Animal Model

Regulation of the expressions of Hif-1α, Vegf, Neurofilaments, NeuN, Thy-1, and Gfap proteins in the rat retina was analyzed by Western blotting at 1, 2, and 4 weeks after optic nerve compression. At 1 week, Thy-1 expression was significantly increased by NPCs and PSCs compared to the age-matched sham group. NPCs significantly induced Vegf in the retina compared to the sham group and PSC group. They also increased Neurofilament induction compared to the sham group at 4 weeks (Figure 3A and Appendix A).

### 2.4. Comparison of the Target Protein Expression between the Retina and the Optic Nerve Tissue in the Optic Nerve Compression Model

We compared the expressions of target protein in the retina versus optic nerve using the ONC model. BDNF expression in the retina was high in the NPC group at 2 weeks, while expression in the optic nerve was high in both the NPC and PSC groups. For Iba1, the lowered expression in the retina had recovered 2 weeks after NPC injection and at 4 weeks after PSC injection (Figure 3B and Appendix A). The expression of the inflammatory protein Nlrp3 was significantly reduced at 1 week and that of Tnf-α was significantly reduced at 2 weeks in the optic nerves of the NPC group (Figure 3C and Appendix A).

### 2.5. Protective Effects of NPCs and PSCs on the Retinal RGCs in the Optic Nerve Compression Animal Model

We assessed the survival rates of RGCs by counting the numbers of RGCs stained with Brn-3a and Tuj1 in rat retinas. After ONC, we found that only NPCs significantly increased Brn-3a and TUJ1 expression more than that in the age-matched sham group in the retina at 4 weeks (Figure 4A,B).

### 2.6. Effects of NPCs and PSCs on Optic Nerve Axon Damage in the Optic Nerve Injury Animal Model

We assessed the protective effects of NPC injection on optic nerves by calculating the GAP43 and Iba1-positive cells from the optic nerve of ONC models, which confirmed that the expression of GAP43 was significantly increased at 2 weeks in both treatment groups, compared to the ONC group (Figure 4C). However, the NPC injection only showed significant recovery up to 4 weeks (Figure 4C). In addition, we figured it out that NPC injection reduced the expression of Iba1 in the optic nerves for 4 weeks, indicating that they could promote microglial enrollment to the retina during the recovery period (Figure 4D).

### 2.7. Wnt/β-Catenin Signal Is Involved during Recovery of Retinal Ganglion Cells by NPCs

In previous report, we suggested that Wnt/β-catenin signaling and Nf-κb protein are involved in the neuroprotection of R28 by MSCs [27,28]. In this study, we also examined whether Wnt/β-catenin signaling may be involved with NPC in the recovery mechanisms of damaged R28 cells. We found that Wnt/β-catenin signaling mediated the NPC induced recovery process. When R28 cells were cocultured with NPC, the CoCl_2_-induced reduction of Wnt3a significantly recovered compared to controls. In addition, the levels of Nf-ĸb expression were maintained similar to normal in the NPC group, and were lower than normal (and the NPC group) in the PSC group (Figure 5).

## 3. Discussion

MSCs are easy to harvest from body fat, bone marrow, placenta, and umbilical cord. In addition, they are immune-privileged because they express low levels of HLA class I antigens and do not present, or present very low levels of, CD80, CD86, CD40, and HLA class II antigens. Furthermore, other unique features such as easy separation, rapid growth after a short period of dormancy, and exemption from ethical issues make MSCs useful for cell therapy [1]. However, NPCs were derived from various conditions and chemical induction in many experiments [26].

The mechanisms by which MSCs directly regulate neuroprotection remain unclear. PSCs have neuroprotective effects via the moderation of Hif-1α and GAP43 [22] and mediation of NF-κb pathways [27,28]. These results suggest that MSC-based therapy may be useful for treating optic nerve disorders, but the critical pathway for optic nerve recovery is unclear. 

In this study, we investigated the mediator proteins and pathways of NPCs associated with the recovery process of the RGC precursor cells. The expressions of Gap43, Thy-1 and neurofilaments, markers of neuronal regeneration, were induced by NPCs. We confirmed the essential functions of NPCs *in*
*vivo* and *in vitro*. The neuroprotective and pro-regenerative effects of NPCs were superior to those of PSCs. To compare the therapeutic roles of PSCs and NPCs, we conducted BDNF-mediated functional analyses. Several previous studies have examined the therapeutic potential of MSCs from various origins.

Previously, we demonstrated the therapeutic efficacy of exosomes from MSCs. They enhanced the neuroprotective and neurorecovery abilities of in vitro models [29]. UBA2 expression is increased in a time-dependent manner in response to hypoxia, and is upregulated in the exosome [1]. Upregulation of Hif-1a/Vegf in MSCs by hypoxic preconditioning has also been reported [1]. Activation of Hif-1a/Vegf signaling and its subsequent involvement in the reduction in apoptosis, autophagy, and inflammation has been reported in hypoxia-preconditioned MSCs [1]. In addition, VEGF induced by MSCs can mediate the differentiation of endothelial progenitor cells via paracrine effects, and anti-VEGF also inhibits the differentiation of endothelial progenitor cells [1].

In the present study, PSCs and NPCs were associated with Iba1 protein expression. NPCs induced microglial expression of Iba1 in the retina more than did PSCs during the recovery period. After optic nerve injury, activated microglia was observed to migrate from the optic nerve to the retina and participate in the cellular response in the damaged retina for the survival and clearance of the axons [30]. The expression of Iba-1, a microglia biomarker, is associated with microglia polarization. During the neuroprotection process, a microglial switch from M1 to the M2 phenotype was induced [31]. Electroacupuncture enhanced the activations of M2 marker Arginase 1 (Arg1) and Iba1-positive cells in the hippocampus, when used as a treatment for neurodegenerative diseases such as Alzheimier’s disease [32]. The induction of Iba-1 expression in retina may contribute to the expression of M2 biomarkers during neuroprotection after optic nerve injury. Thus, NPCs may be more efficient for neuronal rehabilitation after hypoxic injury. Overall, under the hypoxic condition, PSCs and NPCs produced similar effects on the injured RGCs and optic nerve axons. However, considering that the recovery effect of NPCs was more associated with microglial neuroprotection, we would anticipate that NPCs rescue damaged RGCs through cellular interactions better than PSCs. Although further studies are needed to elucidate NPCs’ related pathways, this anti-inflammatory effect is another substantial option to improve the function of MSCs [11], which is relevant to their beneficial effects on various diseases, such as traumatic injuries and neurological disorders. Furthermore, this approach would increase the feasibility of stem cell administration for patients with incurable diseases.

Regarding the routes of stem cell administration, the type of eye disorder would determine how to maximize therapeutic efficiency. For retinal diseases such as retinitis pigmentosa, Stargardt’s disease, and age-related macular degeneration, intravitreal injection would be ideal. For optic nerve disorders such as glaucoma, traumatic optic neuropathy, and neuromyelitis optica, various types of injection have been attempted in clinical trials, including subtenon, intravenous, endonasal, and intrathecal routes [21]. In a previous study that used placental MSCs for traumatic optic neuropathy, a subtenon injection technique was successfully performed and a positive result was reported [33]. In our study, we also injected PSCs or NPCs via the subtenon route, as it is considered less invasive and safer for repeated injections compared to other routes such as intravenous or intravitreal. Regarding the treatment efficacy of NPC injections to treat optic nerve injury, a conventional dose with the same numbers of PSCs maintained a longer-lasting effect. 

A limitation of this study is that we evaluated the recovery function of NPCs and PSCs on injured RGCs and optic nerve axons using only molecular and cellular findings, rather than functional analysis such as a visual evoked potential test. However, the electrical stimulation test would be dependent on molecular recovery and cellular regrowth. Regarding the observation period, 4 weeks may be a relatively short time for assessing regeneration after optic nerve injury. The long-term neuroprotective impact on damaged neuronal tissue should be examined, because a sustainable remedial effect of MSCs would be anticipated clinically. In addition, further studies are needed to verify the mechanisms of neuroprotection related to enhanced functions of NPCs.

In conclusion, NPCs had beneficial effects on hypoxia-injured R28 cells and in an animal model of ONC. NPCs may rescue the damaged RGCs via upregulation of neuroprotection factors, microglial engagement, and anti-inflammatory regulation mediated by Wnt/β-catenin signaling and Nf-ĸb. These cells may be a useful cellular treatment for various optic neuropathies.

## 4. Materials and Methods

### 4.1. In Vitro Study

#### 4.1.1. Human Pluripotent Stem Cell-Derived Neural Progenitor Cells (NPCs) Preparation

##### Culture of Human Pluripotent Stem Cells

H9 human ESCs (WiCell Research Institute, Madison, WI, USA) were routinely maintained on Matrigel-coated culture dishes (BD Biosciences, San Jose, CA, USA) in TeSR™-E8™ medium (STEMCELL Technologies, Vancouver, BC, Canada). For passaging, ESCs were incubated with 0.5 mM EDTA (Thermo Fisher Scientific, Waltham, MA, USA) in a 37 °C CO_2_ incubator for 3 min and then were split in the ratio of 1:20 onto Matrigel-coated dishes with TeSR™-E8™ medium containing 10 μM Y-27632 (Sigma-Aldrich, St. Louis, MO, USA). TeSR™-E8™ medium without Y-27632 was changed daily from two days after passage. Experiments using hESCs were approved by the Institutional Review Board of CHA University (IRB No. 1044308-201603-LR-004-09).

##### Embryoid Body (EB) Formation and Induction of NPCs

Human ESCs were detached with 2 mg/mL collagenase Type IV (Worthington Biochemical Corporation, Lakewood, NJ, USA), for 30 min at 37 °C. EBs were formed from the detached ESCs and were cultured in suspension for 4 days in DMEM/F12 (Thermo Fisher Scientific) containing 10 μg/mL human insulin, 9 μg/mL transferrin, 14 ng/mL selenite, 5 μM PKCβ inhibitor, and 1 μM DMH1 (all from Sigma-Aldrich). The culture medium was changed daily. On Day 4 of culture, EBs were transferred to Matrigel-coated dishes with NPC specification medium containing 1% N2 supplement (Thermo Fisher Scientific), 20 ng/mL bFGF (CHAbiotech, Pangyo, Korea), and 25 μg/mL human insulin (Sigma-Aldrich). The medium was changed every day for 5 days to generate neural rosettes.

##### Flow Cytometry Analysis

Cells were dissociated into single cells with Accutase (Thermo Fisher Scientific) for 5 min at 37 °C and were fixed in 4% paraformaldehyde/phosphate-buffered saline (PBS) for 15 min at room temperature (RT). The fixed cells were treated with 0.2% Triton X-100 (Sigma-Aldrich)/PBS for 15 min at RT and incubated in blocking solution (1% BSA/PBS) for 30 min at RT. The cells were stained with anti-SOX1-PE, anti-PAX6-APC (all from Miltenyi Biotec GmbH, Bergisch Gladbach, Germany) overnight at 4 ℃. Since anti-p75-PE (Miltenyi Biotec GmbH) targeted a cell-surface antigen, p75, the permeabilization process was excluded. The isotype-matched IgG was used as a control. The cells were washed once in 1% BSA/PBS and analyzed using the CytoFLEX Flow Cytometer (Beckman Coulter, Brea, CA, USA).

##### Immunocytochemistry

Cells were fixed in 4% paraformaldehyde/PBS and permeabilized in 0.2% Triton X-100 for 15 min each at RT. The cells were blocked with 5% BSA/PBS for 1 h at RT and were treated with primary antibodies overnight at 4 ℃: The primary antibodies used were targeting TuJ1 (Covance, Burlington, NC, USA) and MAP2 (Millipore, Burlington, MA, USA), and the fluorescence-conjugated secondary antibodies used were Alexa Fluor 488 and Alexa Fluor 594, respectively (all from Thermo Fisher Scientific). The samples were treated with 4′,6-diamidino-2-henylindole (DAPI) (Sigma-Aldrich) for 10 min after the secondary antibody treatment. Images were taken using a ZEISS fluorescence microscope (ZEISS, Oberkochen, Germany).

#### 4.1.2. Human Placenta-Derived Mesenchymal Stem Cells (PSC) Preparation

Human placenta stem cells were collected from Cha General Hospital in Seoul, South Korea. Sampling and use for research purposes were approved by the institutional review committee of the hospital. Preparation and culture operations were performed as previously reported [23].

#### 4.1.3. Mammalian Cell Culture and Treatment

The R28 retinal precursor cells were incubated in Dulbecco’s minimal Eagle’s medium (DMEM; Sigma-Aldrich) with 10% fetal bovine serum (FBS; Thermo Fisher Scientific), 1× minimal essential medium (MEM) with nonessential amino acids (Thermo Fisher Scientific), 100 μg/mL gentamicin (Sigma-Aldrich), and 1% penicillin–streptomycin (Thermo Fisher Scientific). Hypoxic condition was caused by the exposure of R28 cells to cobalt chloride (CoCl_2_) (Sigma-Aldrich). R28 cells (2 × 10^5^) were seeded in a 6-well plate, and NPC or hPSC was co-cultured with R28 cells 3 h prior to CoCl_2_ treatment. Then, R28 cells were treated with CoCl_2_ (300 μM), and the samples were prepared for experiment 24 h later.

#### 4.1.4. Cell Viability Assay

Cells were collected 24 h after co-culturing of NPC or PSC (2 × 10^5^) with hypoxic R28 cells and calculated by microscopy. Cells were stained with trypan blue reagents and only cells identified as surviving cells were counted. The data are presented as the percentage of viable cells (means ± SEMs) in the experimental group compared to the control group.

#### 4.1.5. Immunoblot Analyses

Regenerative and inflammatory markers were analyzed using optic nerve tissue. Lysates were produced from optic nerve tissue using a PRO-PREP solution (iNtRON Biotechnology, Gyeonggido, Korea). The same amounts of total proteins were separated by SDS-electrophoresis and transferred to the membrane. The membranes were incubated with anti-Thy-1 (SC-53116), β-actin (SC-47778) (Santa Cruz Biotechnology, Santa Cruz, CA, USA), Vegf (GTX102643), Tnf-α (GTX10520), β-catenin (GTX101435), Wnt3a (GTX128101), (GeneTex, Irvine, CA, USA), or GFAP (#3670), Neurofilaments (#2837), tCaspase3 (#9662), Bcl2 (#2764), Nf-κb (#8242) (Cell Signaling Technology, Danvers, MA, USA) or Hif-1α (PA1-16601), Bdnf (PA5-85730), Iba1 (PA5-27436) (Thermo Fisher Scientific) or Nlrp3 (NBP2-12446) (Novus Biologicals, Centennial, CO, USA) or NeuN (MABB377) (Millipore) antibodies. All antibodies except Thy-1 (1:200 dilution) were used in a 1:1000 dilution ratio. After washing steps, the membranes were incubated with horseradish peroxidase-conjugated anti-rabbit or mouse secondary antibodies at a 1:10,000 dilution (GeneTex) for o/n at 4 °C. Immuno-active bands were visualized as enhanced chemiluminescence solutions (Bio-Rad Laboratories, Hercules, CA, USA) and detected using ImageQuant™ LAS 4000 (GE Healthcare, Chicago, IL, USA).

### 4.2. In Vivo Study

#### 4.2.1. Animals and the Study Group

Six-week-old male Sprague-Dawley (SD) rats (Orient Bio, Gyeonggido, Korea) were housed in standard animal facilities where food and water were provided at constant temperatures of 21 °C. The in vivo experiment protocol was approved by the Institutional Animal Care and Use Committee of Bundang CHA Medical Center (IACUC200138). The rats were classified into the following groups: Sham (balanced salt solution (BSS) injection after optic nerve compression); NPC group (2 × 10^6^/0.06 mL injection after optic nerve compression); PSC group (2 × 10^6^/0.06 mL injection after optic nerve compression). After 1, 2, and 4 weeks, the animals were euthanized.

#### 4.2.2. The Optic Nerve Compression Model and Subtenon Cell Injection

The rats were anesthetized by Zoletil and Rompun. Animal model production was carried out as mentioned in the previous study [22]. After locally applying 0.5% proparacaine hydrochloride, lateral canthotomy and conjunctival incision were performed. Tissues enclosing the optic nerve were dissected. Using ultra-fine self-closing forceps, optic nerve was compressed at 2 mm site behind the globe for 5 s. Optic nerve compression (ONC) was performed in the left eye (oculus sinister; OS). Then. the canthal incision was sutured. After thorough suturing of the canthal site, subtenon injection of NPCs or PSCs into the nasal side of the eyeballs of the rats was performed.

#### 4.2.3. Assessment of Axon Regeneration Factors in the Optic Nerve of ONC Model

For in vivo measurement of axon regeneration, the vertical portion stained with GAP43 of the optic nerve was photographed. The optic nerves were fixed with 4% paraformaldehyde and embedded in paraffin. The optic nerve was cut vertically to a thickness of 20 μm and mounted on glass slides. Anti-GAP43 antibody (1:200, ab75810; Abcam, Cambridge, UK) or Iba1 (1:200, PA5-27436, Thermo Fisher Scientific) was used to stain the regenerating fibers. The measurement site was a rectangular area of W 150 μm × H 700 μm on both sides of the ONC area, followed by computation of the mean. Total GAP43 or Iba1-positive cells were determined using ZEN software (Carl Zeiss, Jena, Germany).

#### 4.2.4. Flat-Mounted Retinas and RGC Survival Analyses

After the enucleation of three rats from each treatment group, the retina was dissected with a flattened whole mount. After taking out of the cornea by cutting a circular path along the ora serrata with small scissors, the lens was removed using forceps. Separation of the retina from the eyecup was performed by placing forceps between the retina and the eyecup. After obtaining the entire retina, it was cut into quarters using scissors to incise from the retinal to the optic nerve, as described in previous study [1]. The retinas were fixed in 4% paraformaldehyde and mounted on a glass coverslip for at least 1 h at room temperature. After washing with PBS, they were incubated in PBS with 1% Triton X-100 at room temperature for 30 min. The retina was blocked in 20% fetal calf serum for 1 h, and incubated with anti-Tuj1 (ab18207; Abcam) or anti-Brn-3a (MAB1585; Millipore) at a 1:10 dilution for overnight at 4 °C. On the next day, the retina was washed with PBS-T and incubated with goat anti-rabbit IgG-fluorescein isothiocyanate and Alexa Fluor 633 antibodies in PBS-T at 1:200 for 2 h. Before mounting on the coverslip, retina was washed again. Images captured using a confocal microscope (LSM 880; Carl Zeiss, Jena, Germany) were used to quantify fluorescence. Two areas were calculated on each retina, and the average values were compared for statistical analysis.

### 4.3. Statistical Analyses

Data analyses were conducted using GraphPad Prism9 (GraphPad, La Jolla, CA, USA). Statistically significant differences were identified using the *t*-test or nonparametric statistical test, followed by the Mann–Whitney U test at a significance level of 5%.

## Figures and Tables

**Figure 1 ijms-22-12529-f001:**
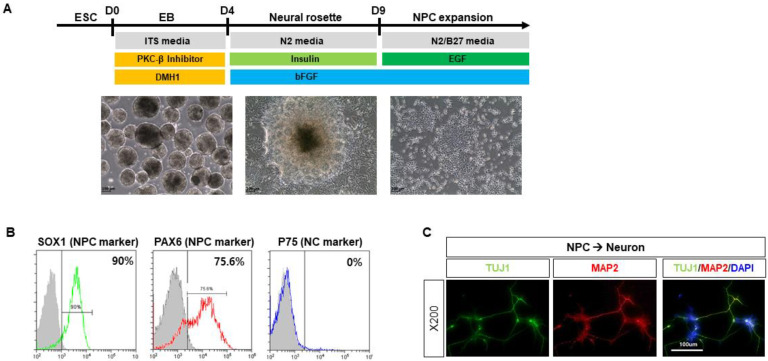
Characterization of human pluripotent stem cell-derived neuronal progenitor cells (NPCs). (**A**) CHA15 human ESCs were differentiated into NPCs by the treatments with 5 μM PKCβ inhibitor, and 1 μM DMH1 in the medium consisting of DMEM/F12, 10 μg/mL human insulin, 9 μg/mL transferrin, and 14 ng/mL selenite. (**B**) Expanded NPCs at passage 1 were shown to be positive for two representative NPC markers, SOX1 (~90%) and PAX6 (~75.6%), but negative for a typical neural crest stem cell marker, P75 (~0%). (**C**) When further differentiated into mature neurons, the NPCs generated both early and late neuronal markers, TUJ1 and MAP2, respectively.

**Figure 2 ijms-22-12529-f002:**
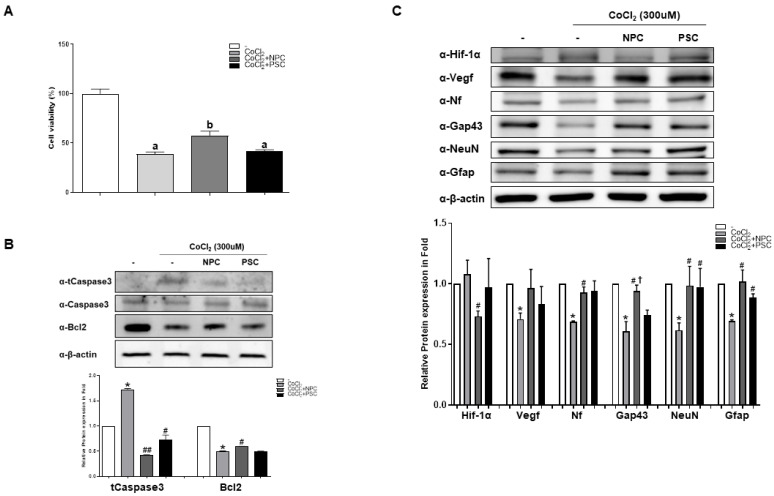
Human NPCs have the function of recovering damaged R28 cells. R28 cells were cultured with NPCs or hPSCs 3 h prior to CoCl_2_ treatment. Then, the R28 cells were treated with CoCl_2_ (300 μM). (**A**) Viability assays performed after 24 h. Data are expressed as percentage (mean ± SEM) of viable cells compared to those in the control group. Significantly different values between groups are indicated by different letters (*p* < 0.05). (**B**) Apoptosis-related protein expressions were also determined. (**C**) Western blot analyses of target protein expression levels, using R28 lysates with CoCl_2_. The quantified values of target protein expression are presented (bottom panel) (* *p* < 0.05 vs. the control; ^#^
*p* < 0.05, ^##^
*p* < 0.01 vs. CoCl_2;_ † *p* < 0.05 vs. PSCs).

**Figure 3 ijms-22-12529-f003:**
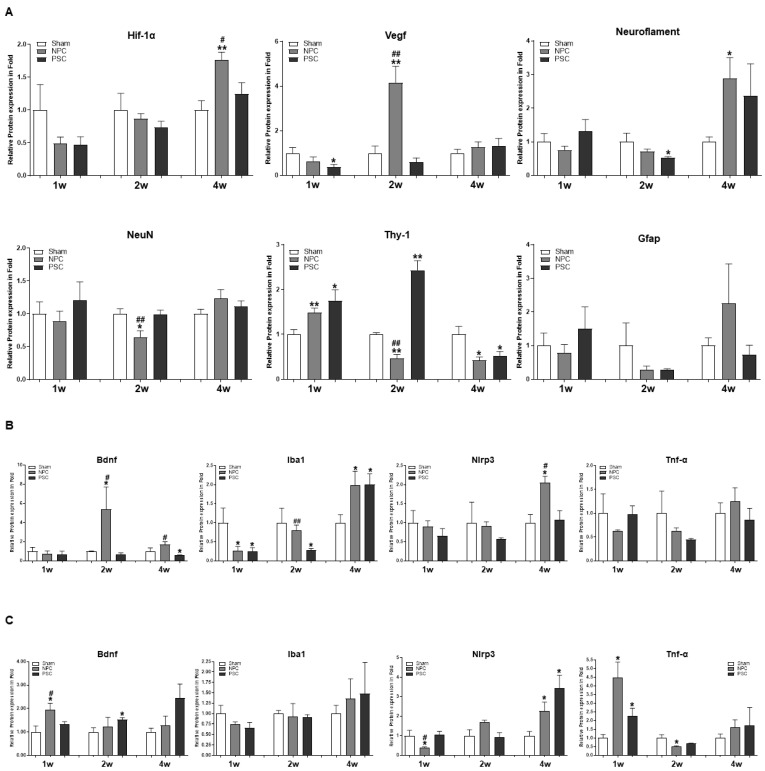
NPCs regulate axonal regeneration and inflammatory proteins in the retina and optic-nerve-injured rat model. Changes in target proteins were assessed by immunoblot analyses of rat retina and optic nerve extracts. The samples were analyzed 1, 2, and 4 weeks after injection with optic nerve compression. Expression levels were normalized to β-actin and the values of OS were divided OD. (**A**) Quantified values of Hif-1α, Vegf, Neuroflament, NeuN, Thy-1and Gfap expressions in retina extract are also presented. Quantified values of Bdnf, Iba1, Nlrp3 and Tnf-α expression in (**B**) retina and (**C**) optic nerve tissues extract are also presented. The results are expressed as the mean ± SEM of independent retina and optic nerve analyses, and are expressed as fold changes compared to the control (* *p* < 0.05 vs. the age-matched sham (balanced salt solution): ^#^
*p* < 0.05; ^##^
*p* < 0.01 vs. PSCs). OD, oculus dexter; OS, oculus sinister.

**Figure 4 ijms-22-12529-f004:**
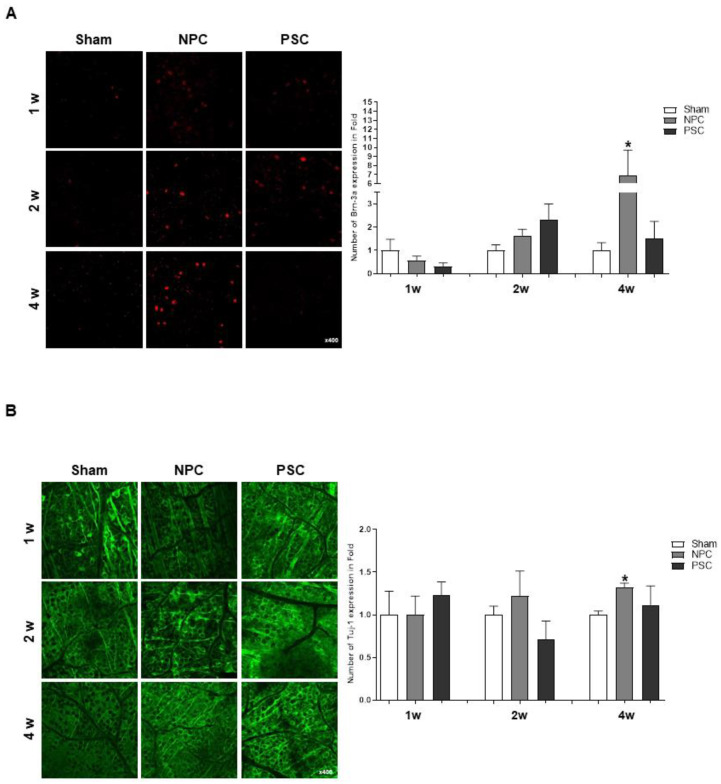
Effects of NPCs promote RGCs and axon regeneration in ONC models. Representative confocal microscopy-based fluorescence images following (**A**) Brn-3a and (**B**) Tuj1 staining (original magnification: 400×) of NPCs and hPSCs injection in the optic nerve compression animal model. (**C**) Gap43 and (**D**) Iba1 fluorescence quantification was performed at a 500 μm distance from the ONC site of the optic nerve. Total GAP43-positive cells were measured using ZEN software. Two retinas and optic nerves from each group were used. The results are presented as the mean ± standard error of the mean (SEM) (* *p* < 0.05 vs. the age-matched sham (balance salt solution): ^#^
*p* < 0.05 vs. PSCs).

**Figure 5 ijms-22-12529-f005:**
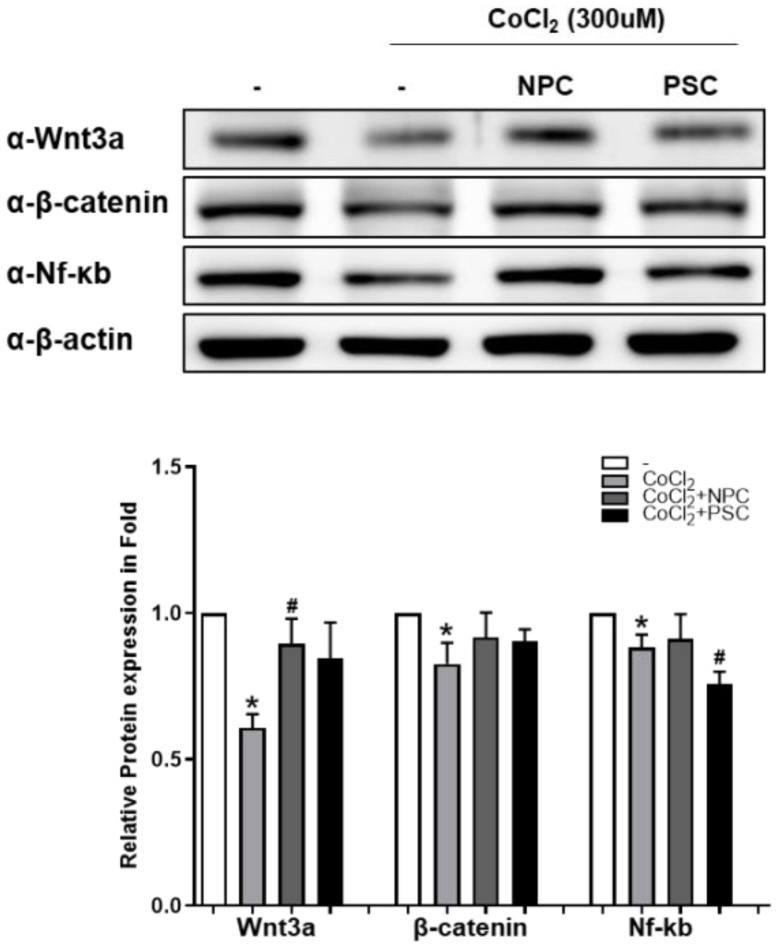
Wnt/β-catenin and Nf-kb during recovery process of damaged R28 cells. After being co-cultured with NPCs or hPSCs, R28 were exposed to CoCl_2_ (300 μM). After incubation for 24 h, Western blotting analyses were performed. The results are expressed as the mean ± standard error of the mean (SEM) (* *p* < 0.05 vs. the control; ^#^
*p* < 0.05 vs. CoCl_2_).

## Data Availability

The data that support the findings of this study are available from the corresponding author upon reasonable request.

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
