# Peer review of "Human Pluripotent Stem Cell-Derived Neural Progenitor Cells Promote Retinal Ganglion Cell Survival and Axon Recovery in an Optic Nerve Compression Animal Model"

_ijms, 2021, doi:10.3390/ijms222212529_

Round 1

Reviewer 1 Report

The study is devoted to an important and unresolved question i.e. improvement of the essential regenerative capacity of retinal ganglion cells. The in vitro and in vivo experiments were correctly designed, performed, presented and discussed. The results suggests that pluripotent stem cell-derived neural progenitor cells can be considered candidates for useful cellular treatment of various optic neuropathies.

Remarks:

Figures 2 and 5: Were the R28 cells separated from NPCs or hPSCs before protein extraction and analysis?

Line 40: RGC, please define the acronym;

Line 106: “in hypoxic conditions by CoCl2”, “in hypoxic conditions simulated by CoCl2”?

Fig. 2B: Densitometric quantification is lacking;

Line 315: “Hypoxic condition was caused by exposure of R28 cells to co-315 balt chloride (CoCl2)”, more precisely, CoCl2 treatment simulate and not cause hypoxic conditions;

Line 327: proteins were separated rather than “decomposed”.

Author Response

Figures 2 and 5: Were the R28 cells separated from NPCs or hPSCs before protein extraction and
analysis?
: The co-culture system was performed using Transwell Membrane inserts. Therefore, R28 cells were not mixed with NPCs or hPSCs. Only R28 cells were collected for protein analysis.
Line 40: RGC, please define the acronym;
: I revised the abbreviation. "Retinal ganglion cells (RGCs)"
Line 106: “in hypoxic conditions by CoCl2”, “in hypoxic conditions simulated by CoCl2”?
: Yes, CoCl2 is commonly used to create a chemically induced hypoxic environment.
Fig. 2B: Densitometric quantification is lacking;
: A quantitative graph was added.
Line 315: “Hypoxic condition was caused by exposure of R28 cells to co-315 balt chloride (CoCl2)”,
more precisely, CoCl2 treatment simulate and not cause hypoxic conditions;
: I agree with you. Numbers of studies have used CoCl2 to induce a hypoxic environment and we checked the expression of Hif-1 or Bnip3, known as a hypoxia marker, after treatment CoCl2.
Line 327: proteins were separated rather than “decomposed”.
: Thanks for the advice, I revised the sentence.

Reviewer 2 Report

This manuscript describes a series of experiments in rat retinal precursor cells and rat retinas, to investigate the effects of neural progenitor cell treatment on retinal ganglion cells and their axons. They use a cell culture model of hypoxia with CoCl2 treatment, and an optic nerve injury model using optic nerve crush injury. For in vitro experiments, co-cultures are used, while for optic nerve crush, subtenon injection of cells is used. Outcomes are Western Blots and immunohistochemistry. Overall the results support the conclusion that NPCs decrease measures of injury.  There are some clarifications that need to be made, though, to improve the manuscript.

- There are some important methodologic details that are missing, which makes some of the data difficult to interpret.

  1. The cell viability assay needs to be described better; is there staining involved to identify non-viable cells? If this is based on morphology, there needs to be a description of the criteria needed.
  2. For animal studies, how many animals per group are used? It appears to be 3, based on the flat-mount retina method, but is this also true for Western analysis?
  3. What are the concentrations of antibodies used? These are specified for a couple antibodies but not for most of them. It might be helpful to just make a table of antibodies used, their source/catalog number and/or RRID, and dilutions used.
  4. For Western Blotting, how are bands quantified? How are they normalized?
  5. For retinal flat mount imaging, how is TUJ1 quantified? It appears that cell counts were used, but it is difficult to count cells with this stain due to how confluent the staining is (e.g. Fig 4B images). Was there a standard approach to counting used?
  6. Similarly, detail is needed on how relative expression of Iba-1 and GAP-43 are measured.

- Results, Page 3 section 2.2: It probably is not correct to use the word apoptosis here, since as far as I can tell no apoptosis markers were used in determining cell viability. There may be cell death, although as noted above there is not enough detail about how cell death assays were done to know whether apoptosis is involved.

- Discussion: more information and discussion is needed regarding the use of CoCl2 treatment as a model of hypoxic injury. It is not entirely clear to me why hypoxia is being used as a cell-culture comparison to trauma, such as was used for the in vivo experiments (this is not really a major concern though). Also, although CoCl2 treatment is used as a model of hypoxia it is important to note (and should likely be discussed as a weakness in the discussion) that CoCl2 does not really induce hypoxia, but mimics some of the cellular effects thereof by stabilizing Hif-1alpha.

- Discussion, page 10, paragraph on lines 220-232: This paragraph sounds like Iba-1 protein expression was increased in stem cell treated groups. This was only true for the 4-week retina samples as measured by Western Blotting, and not in other areas. In fact, Iba-1 expression was decreased at earlier time points, in both the Western Blot analysis and the retinal flat-mount immunohistology. The discussion would benefit from more discussion, including more information about why Iba-1 expression would be expected to be involved in damage recovery. Citations from the literature would be appropriate for this.

- One particular concern I have about the data derived from retinal samples after optic nerve crush is that no data are shown as non-injured controls for the immunoshistochemistry analysis. The authors do attempt to normalize to the non-injured retina (although care needs to be taken with this approach as well. For example, see this paper: https://www.ncbi.nlm.nih.gov/labs/pmc/articles/PMC6888632/. A sham injury control would be very useful for interpreting the data in Figure 4. For example, Fig 4A likely suggests a nearly complete loss of retinal ganglion cells after optic nerve crush (by the photomicrographs under sham injection), with significant improvement in the NPC group. But how much recovery is there compared to the uninjured state?

- Full uncropped Western Blot images are required by the author instructions for this journal, but are not included in this submission.  

- the English usage in the manuscript is sometimes a little difficult to follow, and the paper would benefit from careful proofreading to ensure the meaning is clear. On the whole, though, the meaning of the writing is usually clear.

Author Response

The cell viability assay needs to be described better; is there staining involved to identify non-viable cells? If this is based on morphology, there needs to be a description of the criteria needed.

: I revised the M&M.“ Cells were collected 24 hours after co-culturing of NPC or PSC (2×105) with hypoxic R28 cells and calculated by microscopy. Cells were stained with trypan blue reagents and only cells identified as surviving cells were counted. The data are presented as the percentage of viable cells (means ± SEMs) in the experimental group compared to the control group.”

For animal studies, how many animals per group are used? It appears to be 3, based on the flat-mount retina method, but is this also true for Western analysis?

: As you can see from Supple1, three rat organizations were used for each group in Western analysis. The retina and optic nerve tissue for Western analysis used the same animal.

What are the concentrations of antibodies used? These are specified for a couple antibodies but not for most of them. It might be helpful to just make a table of antibodies used, their source/catalog number and/or RRID, and dilutions used.

: In the method section, the catalog number and company of antibodies were written, and a dilution ratio was added. If necessary, I will make a table of antibodies used.

For Western Blotting, how are bands quantified? How are they normalized?

: The expression of all proteins was quantified with an imageJ program and then normalized with b-actin expression.

For retinal flat mount imaging, how is TUJ1 quantified? It appears that cell counts were used, but it is difficult to count cells with this stain due to how confluent the staining is (e.g. Fig 4B images). Was there a standard approach to counting used?

: As you said, we counted tuj1 stained positive cells. To reduce errors, a dedicated researcher set standards for dyeing and counted them.

Similarly, detail is needed on how relative expression of Iba-1 and GAP-43 are measured.

: The ZEN software is a program dedicated to confocal analysis. It was automatically measured and quantified the amount of expression in the area.

- Results, Page 3 section 2.2: It probably is not correct to use the word apoptosis here, since as far as I can tell no apoptosis markers were used in determining cell viability. There may be cell death, although as noted above there is not enough detail about how cell death assays were done to know whether apoptosis is involved.

: Although not many experiments have been conducted to verify apoptosis, the expression of cleaved case3, known as critical factor of apoptosis, was verified, and it was confirmed that it was significantly reduced by NPC. So I wrote down the apoptosis process.

- Discussion: more information and discussion is needed regarding the use of CoCl2 treatment as a model of hypoxic injury. It is not entirely clear to me why hypoxia is being used as a cell-culture comparison to trauma, such as was used for the in vivo experiments (this is not really a major concern though). Also, although CoCl2 treatment is used as a model of hypoxia it is important to note (and should likely be discussed as a weakness in the discussion) that CoCl2 does not really induce hypoxia, but mimics some of the cellular effects thereof by stabilizing Hif-1alpha.

: It was thought that our optic nerve compression model provided the temporary hypoxic condition by ischemia. In vitro model of hypoxia was established by Cocl2 condition. This condition was verified crucial hypoxic markers (Hif-1a, Bnip3). Although it may not be the same environment as animals, CoCl2, which is widely used to create a hypoxic environment, was used (Sci Rep. 2019; 9: 4898/ Int J Mol Sci. 2021 Aug; 22(16): 8446).

- Discussion, page 10, paragraph on lines 220-232: This paragraph sounds like Iba-1 protein expression was increased in stem cell treated groups. This was only true for the 4-week retina samples as measured by Western Blotting, and not in other areas. In fact, Iba-1 expression was decreased at earlier time points, in both the Western Blot analysis and the retinal flat-mount immunohistology. The discussion would benefit from more discussion, including more information about why Iba-1 expression would be expected to be involved in damage recovery. Citations from the literature would be appropriate for this.

: After optic nerve injury, optic nerve can be a reservoir for activated microglia and other retinal myeloid cells in the retina (https://pubmed.ncbi.nlm.nih.gov/30037353/). Expression of Iba-1, a microglia biomarker, activity is related with microglia polarization. During neuroprotection process, microglial switch from M1 to the M2 phenotype was induced (https://pubmed.ncbi.nlm.nih.gov/32508567/). Treatment for Alzheimier’s disease improved the activation of M2 marker Arginase 1 (Arg1) and Iba1 double positive cells in Hippocampus (https://pubmed.ncbi.nlm.nih.gov/34646113/). Induction of Iba-1 expression in retina may able to activate M2 biomarkers during neuroprotection.

- One particular concern I have about the data derived from retinal samples after optic nerve crush is that no data are shown as non-injured controls for the immunoshistochemistry analysis. The authors do attempt to normalize to the non-injured retina (although care needs to be taken with this approach as well. For example, see this paper: https://www.ncbi.nlm.nih.gov/labs/pmc/articles/PMC6888632/. A sham injury control would be very useful for interpreting the data in Figure 4. For example, Fig 4A likely suggests a nearly complete loss of retinal ganglion cells after optic nerve crush (by the photomicrographs under sham injection), with significant improvement in the NPC group. But how much recovery is there compared to the uninjured state?

: Thank you for your opinion. In the last paper, we derived the results by normalizing the right eye that did not damage (https://pubmed.ncbi.nlm.nih.gov/32524519/). However, the difference on the right side between individuals was so severe that I always thought about setting up the control group. Referring to the RGC subtype analysis profiling study between individuals (https://pubmed.ncbi.nlm.nih.gov/30018341/), we only researched the degree of recovery from damage. Based on the advice you, I will consider setting up the next experiment.

- Full uncropped Western Blot images are required by the author instructions for this journal, but are not included in this submission.  

: I attached Western membrane images.

- the English usage in the manuscript is sometimes a little difficult to follow, and the paper would benefit from careful proofreading to ensure the meaning is clear. On the whole, though, the meaning of the writing is usually clear.

: I requested the editing services to refine it. I will upload the final revised manuscript. Thank you.
